# Structural studies of RFC$^{Ctf18}$ reveal a novel chromatin recruitment role for Dcc1

Benjamin O Wade[1], Hon Wing Liu[2], Catarina P Samora[2], Frank Uhlmann[2] & Martin R Singleton[1],*

## Abstract

Replication factor C complexes load and unload processivity clamps from DNA and are involved in multiple DNA replication and repair pathways. The RFC$^{Ctf18}$ variant complex is required for activation of the intra-S-phase checkpoint at stalled replication forks and aids the establishment of sister chromatid cohesion. Unlike other RFC complexes, RFC$^{Ctf18}$ contains two non-Rfc subunits, Dcc1 and Ctf8. Here, we present the crystal structure of the Dcc1-Ctf8 heterodimer bound to the C-terminus of Ctf18. We find that the C-terminus of Dcc1 contains three-winged helix domains, which bind to both ssDNA and dsDNA. We further show that these domains are required for full recruitment of the complex to chromatin, and correct activation of the replication checkpoint. These findings provide the first structural data on a eukaryotic seven-subunit clamp loader and define a new biochemical activity for Dcc1.

Keywords Ctf18; Dcc1; sister chromatid cohesion; S-phase checkpoint; X-ray crystallography
Subject Categories Cell Cycle; DNA Replication, Repair & Recombination; Structural Biology

## Introduction

Accurate duplication and segregation of chromosomes is vital to maintain integrity of the genetic material. To facilitate this, cells have evolved checkpoint mechanisms to ensure that aberrant processes are detected and where possible, repaired. One such set of checkpoints occur during DNA replication. The so-called intra-S-phase checkpoints are required to maintain genome integrity at stalled replication forks [1–5]. These are characterised by a build-up of single-stranded DNA (ssDNA) that can arise from uncoupled helicase and polymerase activity, resulting from DNA lesions in the template. This ssDNA is recognised by the sensor kinase Mec1 (ATR in humans) which interacts with the ssDNA-binding protein replication protein A (RPA) [6]. Activated Mec1 in turn recruits and activates the diffusible checkpoint kinase Rad53 (Chk2) in conjunction with Mrc1 (Claspin) [7–10]. The activation of Rad53 is associated with its hyper-phosphorylation and induces a range of downstream effects including the stabilisation of stalled forks, repression of late-firing origins [11,12], up-regulation of dNTP synthesis and transcription of DNA repair genes [13–15].

A number of studies in yeast have shown the requirement for another protein complex, RFC$^{Ctf18}$ as being required for full activation of Rad53 [16–18]. RFC$^{Ctf18}$ is a variant replication factor C-like complex, the basic function of which is to load or unload a DNA sliding clamp in an ATP-dependent manner [19,20]. All RFC complexes contain four common subunits (Rfc2-5) and a single variable large subunit that defines the function of the complex [21]. The prototypical member is RFC$^{Rfc1}$, which loads PCNA onto DNA at replication forks and is essential for all replication and repair processes. The best understood variant is probably RFC$^{Rad24}$, which participates in the DNA damage checkpoint response by loading a specialised clamp formed by Rad17, Mec3 and Ddc1 [22–26]. The functions of other variants, RFC$^{Elg1}$ and RFC$^{Ctf18}$ are less clear. RFC$^{Elg1}$ has been implicated in PCNA unloading [27–29] and maintenance of genomic stability [30–36], while RFC$^{Ctf18}$ was originally identified in screens for chromosome mis-segregation [18,37,38]. In addition to containing the Ctf18 variant of the Rfc1 subunit, RFC$^{Ctf18}$ also contains the proteins Dcc1 and Ctf8, making it the only RFC variant formed of seven subunits. RFC$^{Ctf18}$ can both unload and load PCNA *in vitro* [39–41]; however, studies in yeast have suggested that its *in vivo* role is as a loader, as cells lacking Ctf18 show a reduced level of PCNA at sites of DNA replication [42]. The finding that Ctf18 deletion also impinges upon sister chromatid cohesion (SCC) has led to it being classified as a non-essential SCC establishment factor, together with other replication-associated proteins such as Chl1, Ctf4, Csm3, Mrc1 and Tof1 [38,43–47].

Despite the characterisation of the phenotypic effects of RFC$^{Ctf18}$ deletion and its implication in multiple cellular pathways, the biochemical and mechanistic properties of the complex are not fully understood. As the Dcc1-Ctf8 sub-complex is not required for the PCNA-loading or unloading reaction [39,40], we hypothesised that it might play a regulatory role. To investigate this, we have solved the crystal structure of the Dcc1-Ctf8 heterodimer bound to the C-terminus of Ctf18 (Ctf18$^C$). To our surprise, we found that the C-terminus of Dcc1 contains three-winged helix (WH) domains, which we demonstrate can bind to both ssDNA and double-stranded DNA (dsDNA). We further show that these domains aid recruitment

1   Structural Biology of Chromosome Segregation Laboratory, The Francis Crick Institute, London, UK
2   Chromosome Segregation Laboratory, The Francis Crick Institute, London, UK
   *Corresponding author. Tel: +44 203 796 2034; E-mail: martin.singleton@crick.ac.uk

   

of the complex to chromatin *in vivo* and are required for checkpoint activation.

## Results

### Construct design and structure solution

RFC^Ctf18 is formed of seven subunits: Ctf18, Rfc2, Rfc3, Rfc4, Rfc5, Ctf8 and Dcc1 (Fig 1A). The core complex, which is sufficient to load and unload PCNA from DNA [37], is formed of Ctf18 and the four small Rfc subunits. Each of these proteins contains an AAA+ ATPase domain. Previous studies have shown that Dcc1 and Ctf8

are able to form a stable sub-complex *in vitro* and that the C-terminus of Ctf18 is required for the interaction with the Dcc1-Ctf8 sub-complex [39]. To further investigate the interaction between Ctf18 and the Dcc1-Ctf8 sub-complex and the function of Dcc1-Ctf8, we designed a range of constructs using the *S. cerevisiae* proteins for structural studies. We identified a construct that expressed well in *E. coli* that includes the last 76 residues of Ctf18 (residues 666–741) with full-length Dcc1 and Ctf8. Crystals of this complex (designated Ctf18^C-Dcc1-Ctf8) were grown that diffracted to 2.4 Å resolution. While working on this complex, we also obtained crystals of the C-terminal domain of Dcc1 (residues 90–380) that diffracted to 2 Å. Phase information for this crystal form was determined using multiple-wavelength anomalous diffraction

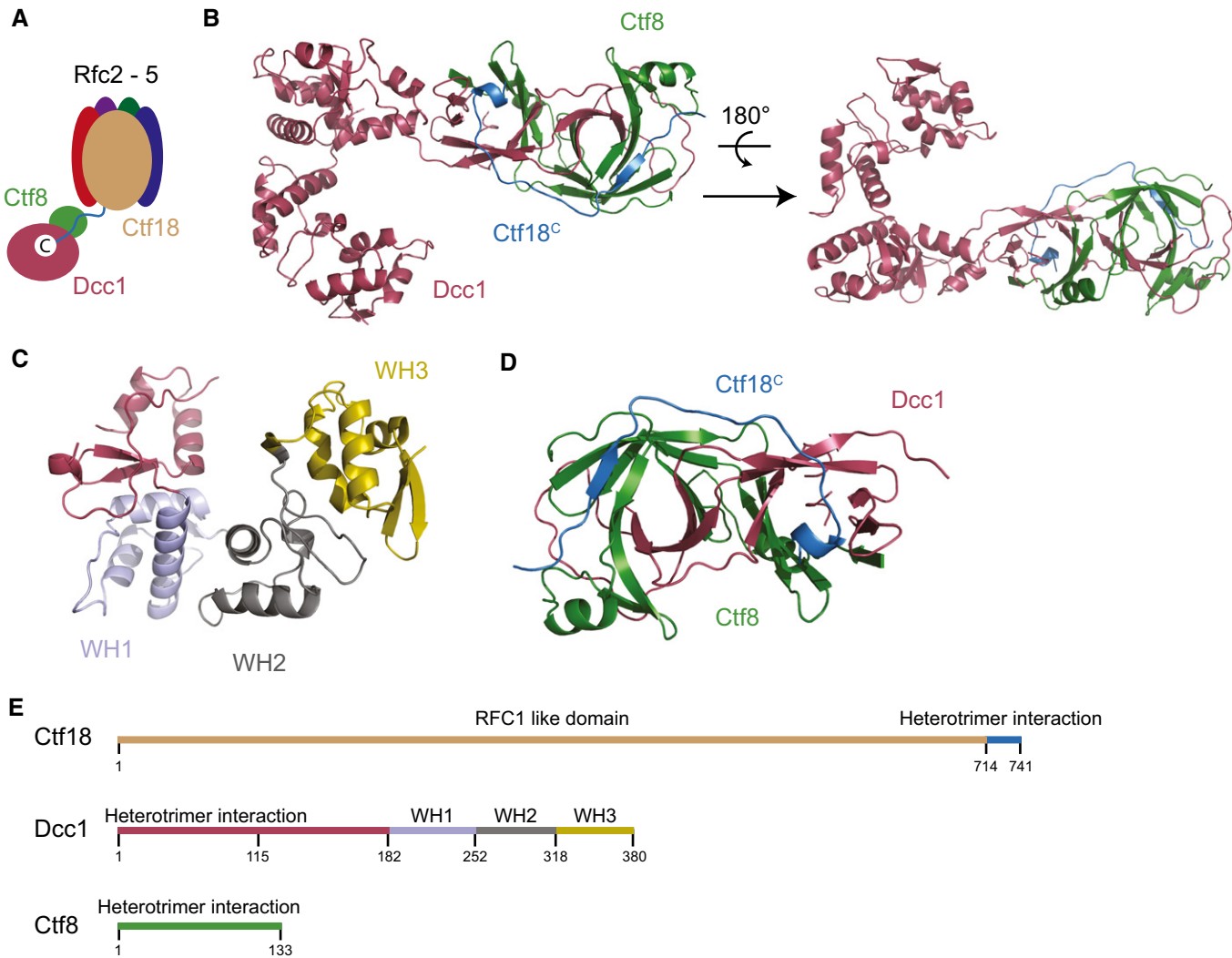

**Figure 1.  The overall structure of the heterotrimer.**

A   Schematic showing the overall organisation of RFC^Ctf18.
B   The structure of the heterotrimeric complex Dcc1 (maroon), Ctf8 (green) and Ctf18^C (blue).
C   Detailed view of the C-terminal domain of Dcc1 with WH1 (blue), WH2 (grey) and WH3 (yellow) identified.
D   The interaction domain between Ctf18 (blue) and the sub-complex, Dcc1 (red) and Ctf8 (green).
E   Linear representation of the overall domain organisation of the heterotrimer.

Data information: See also Figs EV1 and EV2.

(MAD) of selenomethionine-substituted protein. The refined structure was then used as a search model to determine the structure of the Ctf18^C-Dcc1-Ctf8 heterotrimer by molecular replacement (MR). The final structures were of high quality with full statistics of the crystallographic analyses given in Appendix Table S1.

## Overall structure of the Ctf18^C-Dcc1-Ctf8 heterotrimer

The overall structure of Ctf18^C-Dcc1-Ctf8 is extended and looks like a "hook", as the C-terminus of Dcc1 folds back towards Ctf8 (Fig 1B). This C-terminus is primarily α-helical, while the N-terminus

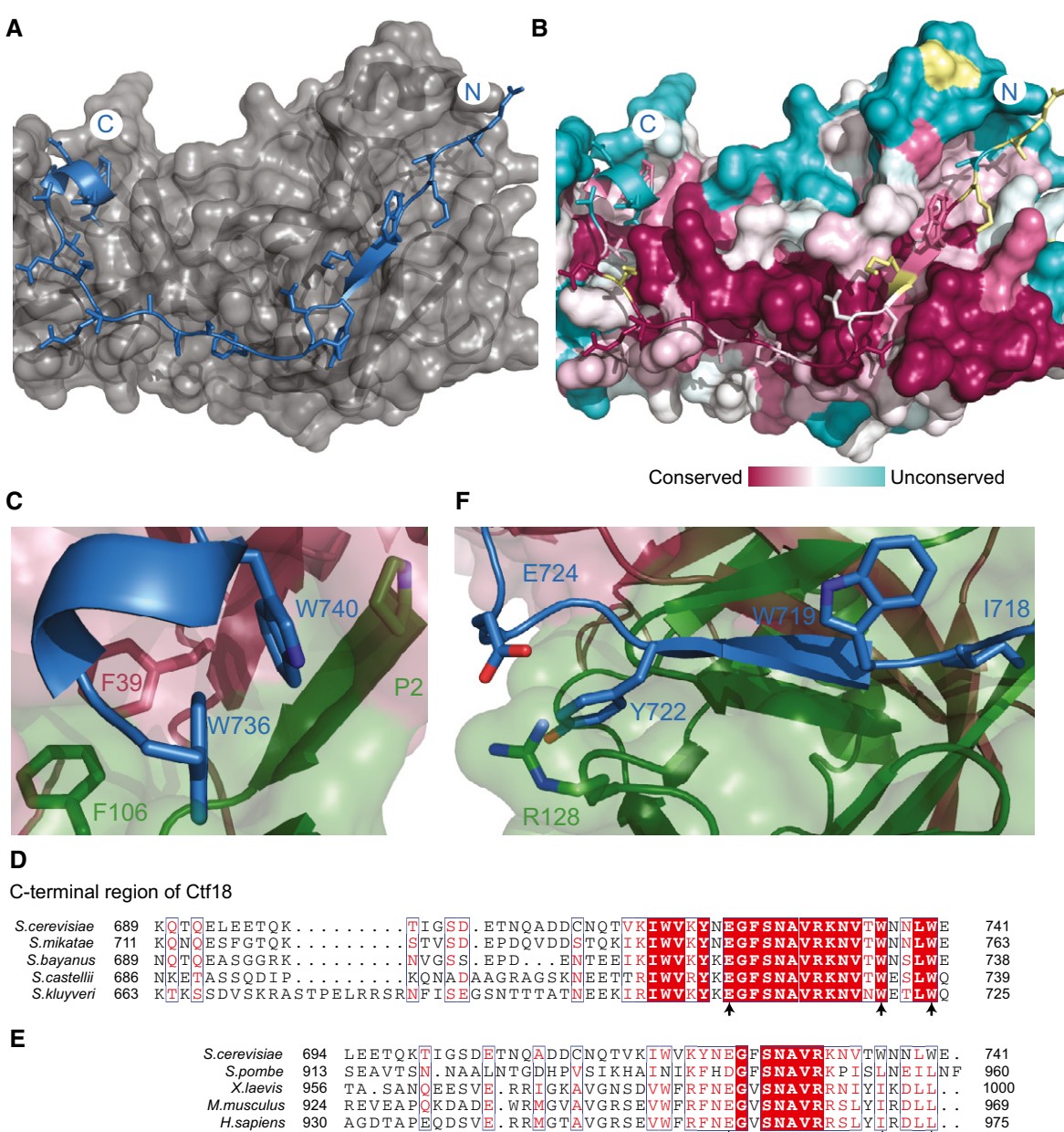

**Figure 2.  Ctf18 binding to the Dcc1-Ctf8 sub-complex.**

A   The binding interface of Ctf18 (blue) residues 714–740 to the surface of Dcc1-Ctf8 (grey).

B   Surface conservation of the Ctf18-binding interface, shown in the same orientation as in (A). Yellow residues indicate that there was insufficient data from the alignment.

C   The C-terminal helix of Ctf18 (blue) with Ctf8 (green) and Dcc1 (maroon). The conserved tryptophans are shown in stick representation.

D   Sequence alignment of yeast species with arrows indicating the location of conserved interacting residues.

E   An expanded sequence alignment from yeast to human showing the functionally equivalent regions. Arrows indicate conserved interacting residues.

F   The interaction of the β-strand region of Ctf18 (blue) with Ctf8 (green) and Dcc1 (maroon).

of the protein and the majority of Ctf8 consist of β-strands. To identify whether any common domains were present in the heterotrimer, we performed a three-dimensional homology search using the Dali server [48]. The results indicated that Dcc1 contains three-winged helix (WH) domains from residues 182–252, 253–317 and 318–380, which we have called WH1, WH2 and WH3, respectively (Fig 1C). These WH domains show structural homology to the WH domains from PKZ [49], cullin-1 [50] and DsrD [51], with Z-scores of 5.7, 7.4 and 4.9, respectively (Fig EV1). WH domains are often

involved in DNA binding or mediating protein–protein interactions [52], suggesting possible functions for this structural feature. The N-terminus of Dcc1 from residues 1 to 115 forms a large heterodimer interface with Ctf8 (Fig 1D). This domain is comprised of 2 α-helices and 18 interconnecting β-strands from both Dcc1 and Ctf8. The overall structure of the heterodimer resembles the "triple" β-barrel fold which has previously been observed in RNase H2 [53], and the transcription factors Rap30/74 [54], Sfc1/7 [55] and A49/34.5 [56] (Fig EV2). This architecture appears to be used

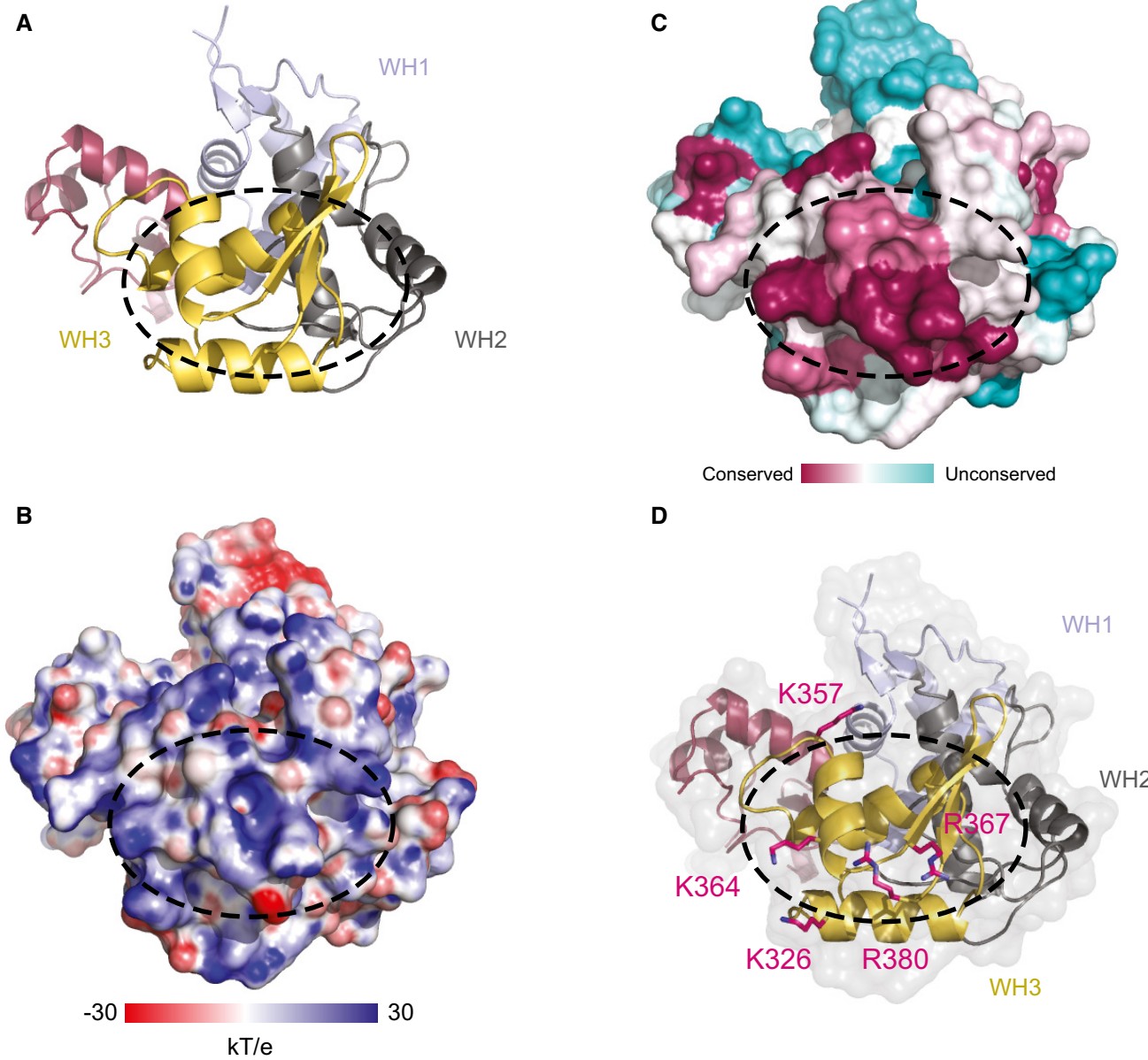

**Figure 3. DNA-binding domain of Dcc1.**

A   Ribbon diagram of the C-terminal domain of Dcc1 with the triple WH domains: WH1 (blue), WH2 (grey) and WH3 (yellow).
B   The surface charge of the C-terminal domain of Dcc1 with positive areas in blue and negative in red.
C   The surface conservation of Dcc1 with highly conserved regions in dark pink and the unconserved regions in pale blue.
D   Potential DNA-binding basic residues tested by mutational analysis depicted in stick representation.

Data information: The dotted areas indicate the presumptive DNA-binding region. Orientation of the structures is the same as in (A).

primarily as a protein interaction motif, linking separate functional domains of larger complexes. The similarity of Dcc1-Ctf8 to the transcription factors listed also extends to sharing a modular structure consisting of a hetero-dimerisation fold preceding tandem WH domains (Fig 1E).

### Ctf18 binds to the Dcc1-Ctf8 heterodimer

Ctf18 binds as an extended peptide to a conserved groove present in the dimer interface of Dcc1-Ctf8 (Fig 2A and B). The extreme C-terminus of Ctf18 forms a short helix, which is partially buried in a hydrophobic pocket of the first "barrel" of Dcc1-Ctf8. The structure shows that W736 and W740 make crucial contacts with other aromatic residues of both Dcc1 and Ctf8 as well as the extreme N-terminal proline of Ctf8 (Fig 2C). These tryptophans are totally conserved in yeast and are always hydrophobic residues in higher eukaryotes, suggesting the overall binding mechanism is conserved (Fig 2D and E). A recent study supports this binding model as co-immunoprecipitation experiments in cells show that mutating both W736 and W740 in Ctf18 disrupt binding to Dcc1-Ctf8 [57]. Furthermore, mutating both residues causes a defect in the phosphorylation of Rad53 in hydroxyurea-treated cells, indicating impaired checkpoint response. The N-terminal section of Ctf18 in our structure forms a beta sheet with Ctf8, with E724 from Ctf18 forming a salt bridge with R128 from Ctf8 (Fig 2F). Residues 666–713 of Ctf18

appear to be unstructured, suggesting they form a flexible linker between the Dcc1-Ctf8 module and the rest of the Rfc1-like domain of Ctf18. The overall binding mechanism is similar to that by which RNase H2A binds to RNase H2B and H2C [53] (Fig EV2) where an extended peptide from H2A binds along the H2B-H2C dimer interface.

### Dcc1 is able to bind DNA through the conserved WH domains

To further investigate the function of Dcc1, we studied both the surface charge and conservation of the WH domains (Fig 3A–D). A large conserved basic patch was present on the surface of WH3. This suggested the domain could interact with DNA as basic residues are able to bind to the negatively charged backbone of DNA. To test this, we carried out an electrophoretic mobility shift assay (EMSA) with fluorescently labelled ssDNA and dsDNA oligonucleotides. The Dcc1 C-terminal construct was titrated against a constant DNA concentration. A gel shift was observed with both ssDNA and dsDNA, suggesting Dcc1 could interact with both substrates (Fig 4A). To further dissect the contribution of the individual WH domains to DNA binding, we deleted WH3 (residue 319–380) from Dcc1 and repeated the EMSA. The results of the initial experiments indicate that WH1 and WH2 could not bind either ssDNA or dsDNA (Fig 4B). However, when we reduced the DNA concentration and overexposed the gel, a slight

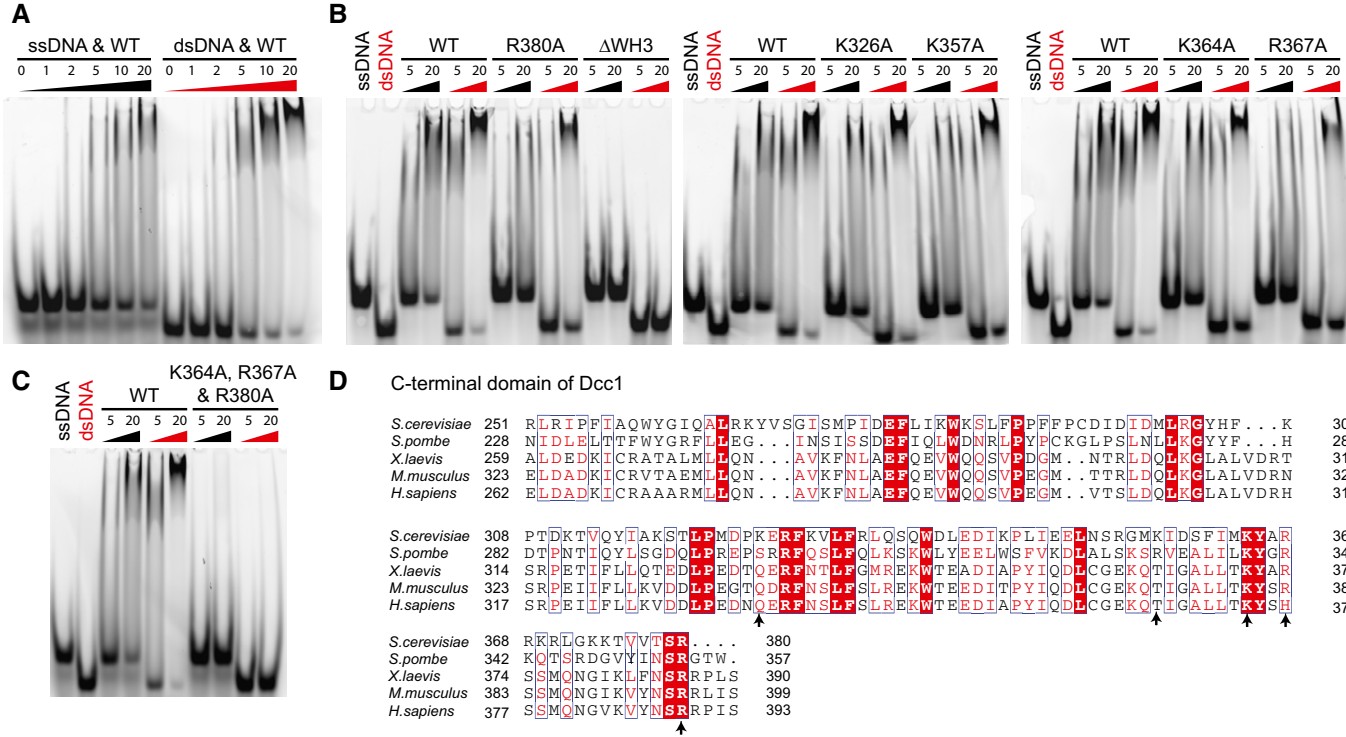

**Figure 4. DNA binding analyses of Dcc1.**

A   DNA binding analysis conducted via an EMSA. 22-mer dsDNA or 25-mer ssDNA was fluorescently labelled, and the C-terminus of Dcc1 (residues 90–380) was titrated in at concentrations indicated. Protein concentrations are given in μM.

B   DNA binding properties of Dcc1 point mutants and WH3 deletion. Protein concentrations are given in μM.

C   DNA binding by triple mutant. Protein concentrations are given in μM.

D   Sequence alignment from yeast to human of WH2 and WH3, with arrows indicating putative basic DNA-binding residues.

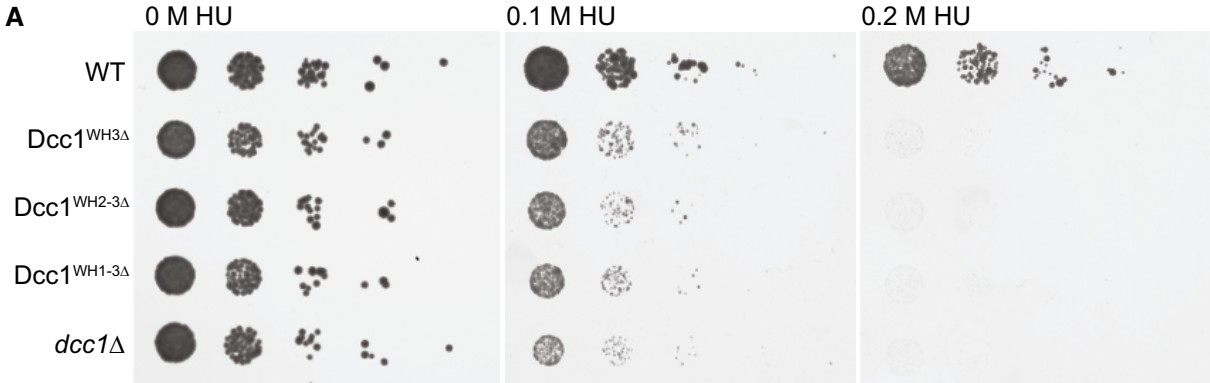

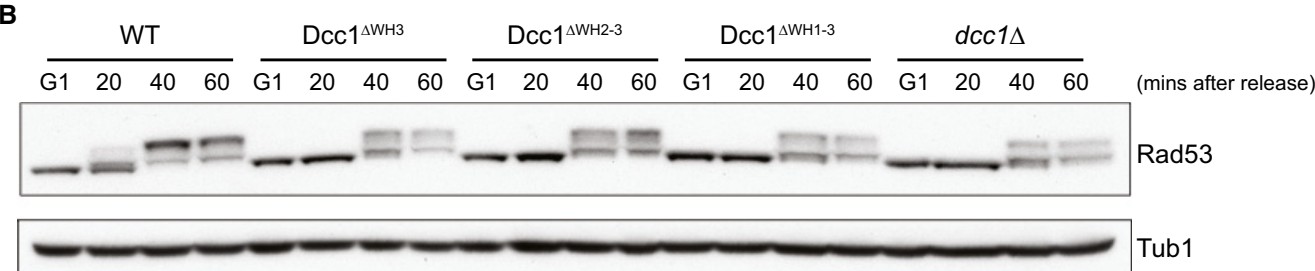

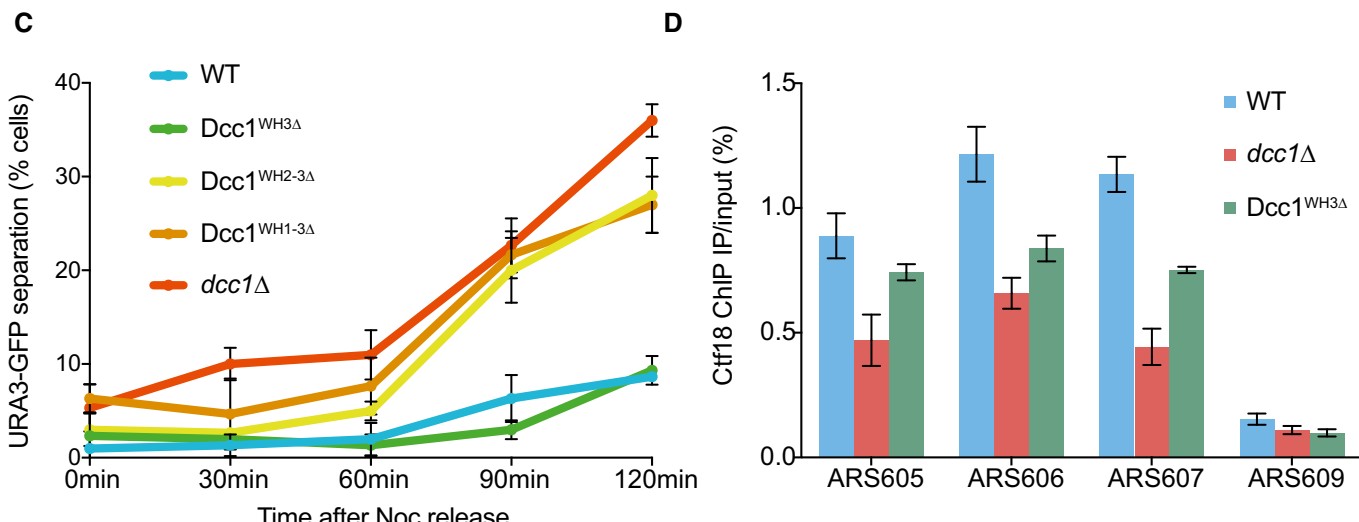

**Figure 5.   Functional analyses of WH domains.**

A   Spot assay showing the effect of deleting WH domains. The resistance of *dcc1Δ*, Dcc1$^{ΔWH3}$, Dcc1$^{ΔWH2-3}$ and Dcc1$^{ΔWH1-3}$ strains compared to wild-type cells in hydroxyurea (HU) was tested by spotting 10-fold serial dilutions on YPD agar containing either 0 M, 0.1 M or 0.2 M HU.

B   Checkpoint activation as assessed by Rad53 phosphorylation. Cells were arrested in G1 and released into medium containing 0.2 M HU for arrest in early S-phase. Samples were taken at the indicated time points and analysed for Rad53 phosphorylation by Western blotting. Tubulin was used as a loading control.

C   Sister chromatid cohesion assay using the same deletions as described above. The effect on sister chromatid cohesion was analysed by visualising tetracycline repressor-GFP fusion proteins bound to tetracycline operator arrays integrated at the *URA3* locus from cells arrested in a nocodazole-imposed mitotic arrest. The percentage of cells with two TetR-GFP dots were recorded. Three independent replicates were performed, and error bars indicate the standard deviation.

D   Analysis of Ctf18 levels at origins in HU-arrested cells. ChIP was conducted with an anti-HA antibody against HA-Ctf18. The enrichment was analysed by qPCR at three early firing origins (ARS605, ARS606 and ARS607) and a late-firing origin (ARS609) on the indicated yeast strains. The means and standard error of three independent experiments are plotted. Analysis of Ctf18 levels averaged over all three early origins by three-way ANOVA demonstrated statistically significant reductions in both the *dcc1Δ* (decrease = −0.555, *P* = 5.06897e-11) and Dcc1$^{WH3Δ}$ (decrease = −0.3018, *P* = 1.1074164e-06) strains.

gel shift was observed with ssDNA but not dsDNA, indicating that WH1 and WH2 have a weak affinity for ssDNA, which is enhanced by the presence of WH3 (Fig EV3A). Dcc1 was also able to bind to dsDNA with a ssDNA extension at either the 3′ or 5′ end with an approximately equal affinity to single-stranded or duplex DNA (Fig EV3B). To identify individual residues involved in binding, we mutated basic residues in WH3 and compared them to wild-type (WT) protein. The residues K326, K357, K364, R367 and R380 (Fig 3D) were mutated to alanine and their effect on DNA binding assessed (Fig 4B). Mutation of K364, R367 or R380 reduced the affinity for both ssDNA and dsDNA, albeit weakly with K364, while the triple mutant K364A, R367A, R380A totally eliminated binding similar to the WH3 deletion (Fig 4C). These residues are widely conserved (Fig 4D) and suggest that the binding site for ssDNA and dsDNA overlaps to some extent.

## Functional studies on the Dcc1 WH domains

Cells lacking Ctf18, Dcc1 or Ctf8 grown in the presence of hydroxyurea (HU) cannot properly activate the intra-S-phase checkpoint at stalled forks [16–18,58]. To further understand the function of the WH domains in Dcc1, we progressively deleted the WH domains and assessed checkpoint activation as assessed by Rad53 phosphorylation as well as viability when treated with hydroxyurea. We found that removal of any of the WH domains caused severe viability problems under HU treatment and showed delayed and reduced Rad53 phosphorylation, comparable to a complete Dcc1 deletion (Figs 5A and B, and EV4A). These defects are unlikely to be caused by impaired heterotrimer formation as Ctf18 and Ctf8 are still able to form a stable soluble complex with the WH3 or WH2-WH3 truncated Dcc1 proteins (Fig EV4B). The WH1-WH3 deletion is insoluble when recombinantly expressed but stable Dcc1$^{\Delta WH1-WH3}$ can be observed in cell extracts (Fig EV4A) suggesting it is also capable of binding Ctf8 and Ctf18, as free Dcc1 is quite unstable and rapidly degraded (data not shown).

We also find that deletion of the WH domains leads to sister chromatid cohesion defects (Fig 5C). Interestingly, however, while deletion of any of the WH domains had a severe checkpoint phenotype, deletion of the WH3 domain alone had a minimal impact on sister chromatid cohesion. We further analysed the function of the WH3 domain by chromatin immunoprecipitation (ChIP) and qPCR on Ctf18 at sites of replication origins in hydroxyurea-treated cells (Figs 5D and EV4C). Cells without the Dcc1 WH3 domain showed a statistically significant reduction of Ctf18 at replication origins, though not as much as complete Dcc1 deletions (Fig 5D). As Ctf18 deletion does not compromise entry into S-phase [16], and the truncated complex is still stable (Fig EV4B), this result suggests impaired recruitment of the RFC$^{Ctf18}$ complex to stalled forks.

## Discussion

RFC$^{Ctf18}$ is a seven-subunit complex formed of Ctf18, Rfc2-5 and the stable Dcc1-Ctf8 sub-complex. The complex is required for activation of the intra-S-phase checkpoint and proper establishment of sister chromatid cohesion. To better understand the

function of RFC$^{Ctf18}$, we determined the crystal structure of the heterotrimeric Ctf18$^C$-Dcc1-Ctf8 complex and analysed *in vitro* and *in vivo* activities.

The structure of Ctf18$^C$-Dcc1-Ctf8 was completely unexpected and revealed that Dcc1 has three consecutive WH domains at its C-terminus, the third of which can bind to both ssDNA and dsDNA. To our knowledge, this is a completely novel organisation of WH domains. The N-terminus of Dcc1 and Ctf8 form a "triple" β-barrel fold, which can then bind Ctf18. This overall architecture has previously been found in a range of transcription factors and RNase H2. Like the transcription factors, RFC$^{Ctf18}$ uses this domain as a bridge between a larger complex and a variable number of WH domains able to bind DNA. This suggests a common ancestor of both a subset of transcription factors and RFC$^{Ctf18}$. RFC complexes are able to directly recognise and bind DNA structures such as template-primer junctions via multiple Rfc subunits [21,59]. The discovery that RFC$^{Ctf18}$ contains additional binding sites for both dsDNA and ssDNA suggests that directed recruitment is an important part of its mechanism, as the intrinsic DNA binding activity of the core complex is sufficient for the basic loading and unloading activity [39,40].

Recent studies have also shown that RFC$^{Ctf18}$ is able to bind to DNA polymerase epsilon (Pol ε) via the catalytic subunit Pol2 [57,60,61]. This interaction requires the intact Ctf18$^C$-Ctf8-Dcc1 complex which we present in this study. Our structure shows that mutations in, or deletion of the Ctf18 C-terminus would dissociate the Dcc1-Ctf8 heterodimer from the complex, so disrupting the platform for Pol ε binding. The addition of two non-Rfc subunits to the RFC$^{Ctf18}$ complex is reminiscent of the bacterial clamp loader, which employs two auxiliary subunits, χ and ψ to link the loader to single-stranded DNA-binding protein (SSB) upstream of the replication fork [62,63]. Dcc1 and Ctf8 bear no structural similarity to χ and ψ [64]. However, the structure of the bacterial clamp loader bound to a peptide derived from the ψ protein [65] as well as the location of the C-terminus of Rfc1 in the RFC-PCNA structure [66] suggests that the linker region of Ctf18 emerges adjacent to the "collar" domain of RFC. This would allow the Dcc1 subunit to interact with both ssDNA emerging from the clamp loader as well as Pol ε, situated on the same face of the complex (Fig 6).

Our structure shows that the Ctf8-Dcc1 interface acts as a binding site for the Ctf18 C-terminus, which together forms a platform capable of binding Pol 2. Given the proximity of the WH domains to this binding site, it seems likely that they are also capable of interacting with the Pol ε complex, and/or DNA within the replisome. The reduced occupancy of Ctf18 on chromatin observed following WH3 deletion supports this notion. Although all the WH domains are required for full checkpoint activation, we find that only WH1 and WH2 are required for correct sister chromatid cohesion. This separation of function might be explained if the cohesion pathway is more tolerant to reduced levels of chromatin bound Ctf18 relative to the checkpoint pathway. Alternatively, the WH3 domain might form a checkpoint-specific interaction with other proteins.

It has been reported that RFC$^{Ctf18}$ also acts in additional pathways, for example telomere positioning and stability [67–69] and opens up the intriguing possibility that Dcc1 has other, as yet undetermined activities that require robust DNA interactions. Determining the precise details of exactly how and where this multi-faceted protein complex contributes to DNA replication and chromosome segregation activities will provide an interesting ongoing challenge.

                                    

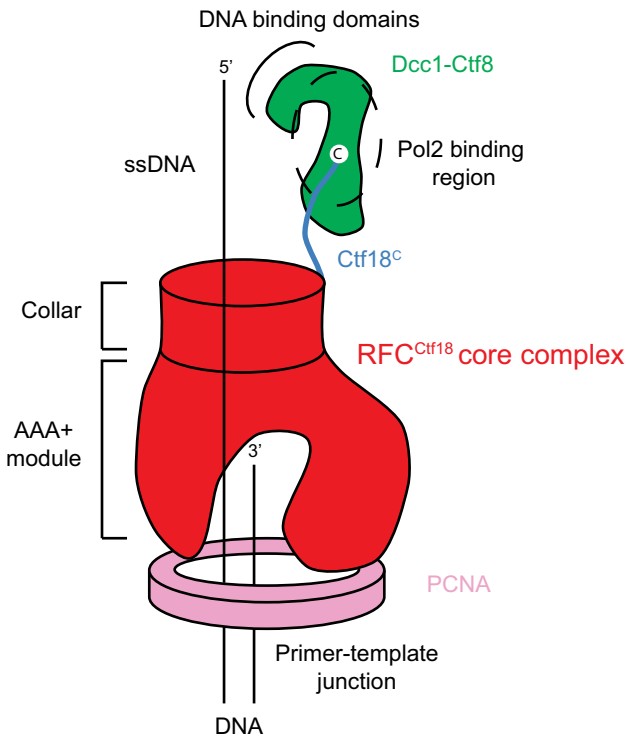

**DNA binding domains**

Dcc1-Ctf8

5′

ssDNA

**C**

Pol2 binding region

Ctf18<sup>C</sup>

Collar

RFC<sup>Ctf18</sup> core complex

AAA+ module

3′

PCNA

Primer-template junction

DNA

**Figure 6. Organisation of the RFC<sup>Ctf18</sup> complex.**
Schematic diagram showing the location of the Ctf8-Dcc1 heterodimer in relation to the clamp loader/clamp and likely location of the Pol ε binding site.

# Materials and Methods

### Protein expression and purification

The genes coding for the subunits Ctf18 (residues 666–741), Dcc1 (full length) and Ctf8 (full length) were amplified from genomic DNA and cloned into the vectors pET-28a, pCDFDuet and pETDuet, respectively. The Ctf18 subunit had the addition of an N-terminal His₆ tag and TEV protease site. These subunits were then sequenced to confirm that no mutations had been introduced. The vectors containing the subunits were transformed into BL21(DE3)-RIL competent cells (Agilent) using standard procedures. An overnight culture of freshly transformed cells was grown at 37°C in LB, which was used to inoculate a 1 l culture. These cells were then grown at 37°C until an OD of 0.6 was reached, upon which expression of the complex was induced by the addition of 0.25 mM IPTG at 16°C for 18 h. The cells were harvested and sonicated in 50 mM Tris pH 7.5, 300 mM NaCl, 10% glycerol and 0.5 mM TCEP and then centrifuged at 35,000 × *g*, 4°C for 1 h. The supernatant was loaded onto a 5 ml His-Trap HP (GE Healthcare) and eluted using imidazole. The tag was then cleaved from the Ctf18 subunit by the addition of TEV protease at 0.04 mg/ml and left overnight at 4°C. The protein was loaded onto a MonoQ 10/300 GL (GE Healthcare) column equilibrated in 20 mM Tris pH 7.5, 100 mM NaCl and 0.5 mM TCEP. The protein was eluted in a high concentration of NaCl. The complex was concentrated and loaded on an equilibrated HiLoad 16/60 Superdex 75 column (GE Healthcare) in 10 mM Tris pH 7.5,

150 mM NaCl and 0.5 mM TCEP. Eluted protein-containing fractions were analysed for purity by SDS–PAGE. Selenomethionine incorporated protein was obtained by growing the cells in minimal media where selenomethionine was the only source of methionine and purification of the protein was identical to the native protein. The Dcc1 truncation (residues 90–380) was expressed and purified using the same protocol as for the full complex.

### Crystallisation and structure solution

The selenomethionine incorporated C-terminus of Dcc1 was concentrated to 20 mg/ml and crystallised in 2 µl sitting drops with equal volumes of protein to reservoir solution at 20°C. The reservoir contained 0.2 M potassium sodium tartrate and 18% PEG 3350. Crystals were harvested after 2 days in 20% ethylene glycol. A two wavelength MAD data set was collected at Diamond Light Source on beamline IO3. The AutoSHARP [70] package was used to solve the structure, and Buccaneer [71] was used to trace the initial model. Iterative rounds of rebuilding and refinement were carried out in Coot [72] and Refmac5 [73]. The final model was then fitted by molecular replacement to a higher resolution native data set obtained from a different crystal form grown in the same conditions. Ctf18<sup>C</sup>-Dcc1-Ctf8 was concentrated to 35 mg/ml and was crystallised in 4 µl sitting drops with equal volumes of protein to reservoir solution. The reservoir contained 0.1 M Bis–Tris propane pH 6.3, 0.2 M NaBr and 17% PEG 3350. Crystals were grown at 20°C for 10 days before harvesting in 15% ethylene glycol and stored in liquid nitrogen. Native data were collected at Diamond Light Source on beamline IO3. Molecular replacement was carried out in Phaser [74] using the C-terminus of Dcc1 as a start model. An initial model was traced using phenix.autobuild [75], and iterative rounds of rebuilding and refinement were carried out as before. Some spots fell beyond the edge of the detector accounting for reduced completeness in the outer shell. Full statistics for data collection and refinement are shown in Appendix Table S1.

### DNA binding analysis

DNA binding analysis was conducted on both dsDNA and ssDNA. The dsDNA had the sequence TCTCCACAGGAAACGGAGGGGT and a fluorescent 6-FAM tag on the 5′ sense strand. The DNA was annealed and purified using standard techniques. The ssDNA used was a 25-mer oligo d(T), with a 5′ 6-FAM fluorescent tag. Both oligos were synthesised and purchased from Sigma-Aldrich. A 6% non-denaturing DNA retardation gel was pre-run at 4°C at 3 mA for 1 h in 0.5× TBE. DNA was at 1 µM unless otherwise stated and protein titrated at increasing concentrations. Samples were loaded and run for 75 min at 5 mA, 4°C. The gel was then analysed using a Typhoon 9400 imager excited at 488 nm, and emission was detected at 520 nm at 500 V.

### Yeast strains and cultures

All strains used in this study were of W303 background and are listed in Table 1. Gene deletions and epitope tagging of endogenous genes were performed by gene targeting using polymerase chain reaction (PCR) products [76,77]. Cells were grown in rich YP medium supplemented with 2% glucose as the carbon source [78]. Synchronisation

**Table 1.   Yeast strains used in this study.**

| |
|---|
| Y141 W303 wild type |
| Y4950 *MATa dcc1Δ::LEU2* |
| Y4951 *MATa dcc1(WH3Δ)-Pk₉::TRP1* |
| Y4325 *MATa ctf18Δ::TRP1* |
| Y1235 *MATa CTF18-HA₆::HIS3* |
| Y4952 *MATa dcc1Δ::LEU CTF18-HA6::KAN<sup>R</sup>* |
| K7100 *MATa URA3::tetOs HIS3::tetR-GFP* |
| Y4954 *MATa ura3::3xURA3 tetO112, his3::HIS3 tetR-GFP dcc1Δ::LEU2* |
| Y4955 *MATa ura3::3xURA3 tetO112, his3::HIS3 tetR-GFP dcc1(WH3Δ)-Pk₉::TRP1* |
| Y5202 *MATa ura3::3xURA3 tetO112, his3::HIS3 tetR-GFP DCC1-Pk₉::TRP1* |
| Y5203 *MATa ura3::3xURA3 tetO112, his3::HIS3 tetR-GFP dcc1(WH2-3Δ)-Pk₉:: TRP1* |
| Y5204 *MATa ura3::3xURA3 tetO112, his3::HIS3 tetR-GFP dcc1(WH1-3Δ)-Pk₉:: TRP1* |
| Y5245 *MATa CTF18-HA₆::HIS3 DCC1-Pk₉::TRP1* |
| Y5246 *MATa CTF18-HA₆::HIS3 dcc1(WH3Δ)-Pk₉::TRP1* |
| Y5247 *MATa CTF18-HA₆::HIS3 dcc1(WH2-3Δ)-Pk₉::TRP1* |
| Y5248 *MATa CTF18-HA₆::HIS3 dcc1(WH1-3Δ)-Pk₉::TRP1* |

in G1 was performed by addition of α-factor (0.4 μg/ml) for 2 h. To arrest cells in early S-phase, G1 synchronised cultures were filtered, washed and re-suspended in fresh medium containing 0.2 M hydroxyurea for 60 min. Arrest in G2/M was achieved by release into medium containing 5 μg/ml nocodazole for 120 min.

**Yeast molecular biology techniques**

Chromatin immunoprecipitation was performed as previously described [79], as was the quantitative analysis of chromatin immunoprecipitates [80]. Analysis of sister chromatid cohesion was done by visualising tetracycline repressor-GFP fusion proteins bound to tetracycline operator arrays integrated at the *URA3* locus on chromosome 5, as previously described [81]. Hydroxyurea (HU) resistance was tested by spotting 10-fold serial dilutions of cultured strains on YPD agar containing 0 M, 0.1 M or 0.2 M HU. Antibodies used for Western blotting or chromatin immunoprecipitation were anti-HA (clone F-7, Santa Cruz), anti-tubulin (clone YOL1/34, Serotec) and an antibody specific to Rad53 (ab104232, Abcam).

**Data availability**

Coordinates and structure factors of Ctf18<sup>C</sup>-Ctf8-Dcc1 and Dcc1<sup>90–380</sup> structures have been deposited in the Protein Data Bank under accession codes 5MSM and 5MSN, respectively.

**Expanded View** for this article is available online.

## Acknowledgements

We wish to thank A. Purkiss for assistance with X-ray data collection, the Crick Proteomics platform for mass spectrometric analyses and G. Kelly for assistance with data analysis. This work was supported by the Francis Crick Institute, which receives its core funding from Cancer Research UK (FC001155, FC001198), the Medical Research Council (FC001155, FC001198) and the Wellcome Trust (FC001155, FC001198).

## Author contributions

BOW cloned, expressed, purified and crystallised all constructs and carried out DNA binding experiments. BOW and MRS collected and analysed X-ray data. CPS and HWL carried out *in vivo* experiments. BOW, FU and MRS conceived and designed the study. All authors provided input into the manuscript.

## Conflict of interest

The authors declare that they have no conflict of interest.

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
