## [Review Process File · EMBO Reports]

Manuscript EMBO-2016-42825

Structural Studies of RFCt18 Reveal a Novel Chromatin Recruitment Role for Dcc1

Benjamin O. Wade, Hon Wing Liu, Catarina P. Samora, Frank Uhlmann, and Martin R. Singleton

Corresponding author: Martin R. Singleton, The Francis Crick Institute

Review timeline:

Submission Date:	01 September 2016
Editorial Decision:	04 October 2016
Revision Received:	14 December 2016
Editorial Decision:	04 January 2017
Revision Received:	05 January 2017
Accepted:	09 January 2017

Transaction Report:

1st Editorial Decision

04 October 2016

Thank you for the submission of your research manuscript to our journal. We have now received the full set of referee reports that is copied below.

As you will see, both referees acknowledge the potential interest of the findings. It appears that most concerns can be addressed with textual changes, however, referee 2 also points out a couple of missing control experiments and indicates that the CHIP assays should be quantified.

Given these constructive comments, we would like to invite you to revise your manuscript with the understanding that the referee concerns (as detailed above and in their reports) must be fully addressed and their suggestions taken on board. Please address all referee concerns in a complete point-by-point response. Acceptance of the manuscript will depend on a positive outcome of a second round of review. It is EMBO reports policy to allow a single round of revision only and acceptance or rejection of the manuscript will therefore depend on the completeness of your responses included in the next, final version of the manuscript.

Revised manuscripts should be submitted within three months of a request for revision; they will otherwise be treated as new submissions. Please contact us if a 3-months time frame is not sufficient for the revisions so that we can discuss the revisions further.

REFeree REPORTS

Referee #1:

To gain further insight into the role of Ctf18-RFC in different cellular pathways, the authors solved the structure of the two accessory subunits Ctf8 and Dcc1 complexed with a C-terminal fragment of Ctf18. The authors found that the C-terminal domain of Ctf18 interacts through conserved residues with both subunits, Ctf8 and Dcc1. Moreover, they identify three conserved WH domains in Dcc1 that bind DNA. A major success of this work was the subsequent characterization of DNA binding: through EMSA experiments the authors demonstrated that Dcc1 by itself is able to bind to ds and ssDNA through highly conserved residues. It was further demonstrated that loss of Dcc1 or its C-terminal WH domain impaired recruitment of Ctf18-RFC to origins of replication. These findings are novel and important to the field as they bring up interesting questions about the physiological importance of Dcc1 and the mechanism of how Ctf18-RFC recognizes DNA.

The manuscript is easy to read and written in a clear language, and represents a significant advance. I recommend publication in EMBO reports.

I do have a few minor critiques, which should be clarified or fixed before publication.

- 1) On page 9, the authors state "Dcc1 has three consecutive WH domains at its C-terminus which can bind to both ssDNA and dsDNA." However, the in vitro data only show a role for WH3 in direct DNA binding. Although WH1&2 are important for efficient recruitment to chromatin at replication forks, this could be through other means than direct binding to DNA. The above statement needs clarification.
- 2) Also on page 9, the authors state "Like the transcription factors, RFC-Ctf18 uses this domain as a bridge between a large complex and WH domains able to bind DNA." But they also claim that "To our knowledge, this is a completely novel organization of WH domains." These two statements seem to be in conflict with each other.
- 3) On page 10: "which is subsequently capable of binding Pol2." This implies an sequential order of binding, but there is no data to support this. This statement needs to be clarified.
- 4) Figure 6 shows the collar region of the clamp loader as an open ring, but it is a closed disk in all known clamp loaders (Bowman et al 2006, Simonetta et al 2009, Kelch et al 2011).
- 5) The nomenclature for the C-terminal fragment of Ctf18 is not consistent throughout the ms.
- 6) No figure calls for Fig. 3B and C.
- 7) The K364A mutation appears to have a rather weak effect on binding and it is not clear whether this should be included in the set of mutations that inhibit binding.

It is this reviewer's policy to review non-anonymously when possible. (I don't want to write reviews that I would feel embarrassed about my tone or content if the authors were to actually know my identity.) If the editor and/or authors want clarification about these reviews, please contact: Brian Kelch PhD, UMass Medical School (brian.kelch@umassmed.edu)

Referee #2:

The manuscript by Wade et al. describes the first structure of the Dcc1 and Ctf8 subunits of the RFC-Ctf18 complex. There are four RFC complexes in eukaryotic cells that serve as clamp loaders that load (and unload) ring-shaped sliding clamps on DNA. These RFC complexes contain four "small" subunits in common, but differ in the large subunit, which gives each RFC complex a different cellular function. As a group, the RFC complexes are essential to DNA replication, maintaining genome integrity, sister chromatid cohesion, and activating cellular checkpoints. RFC-Ctf18 differs from the other RFC complexes in that it contains two small accessory subunits, Dcc1 and Ctf8. Given the critical importance of the RFC complexes to genome maintenance and the many unanswered questions regarding how the differences in subunit composition give the RFC

complexes different cellular functions, this manuscript is highly significant.

Surprisingly, the structure revealed that Dcc1 contains three winged-helix domains that often function as DNA binding and/or protein interaction domains. Dcc1 and Ctf8 also interact through a domain that is reminiscent of RNase H2 and a number of transcription factors. These structural features suggest the Dcc1 protein may be important for DNA binding and localizing RFC-Ctf18 to chromatin for its function. The authors perform experiments to test these hypotheses. These functional assays add another dimension to the paper and increase the impact of their structural data.

Because the biochemical and cellular experiments presented to tease out the functions of Dcc1 are important to this paper, there are a couple of key controls that are missing and need to be done to properly interpret the results. First, the authors must show that the Dcc1 deletion mutants are expressed at levels similar to full-length Dcc1. This is important for all of the experiments in Figure 5 (measuring ChIP relative to input in Fig. 5D does not address this). Second, the authors should show that Dcc1 deletion mutants are assembled into the Rfc1-Ctf18 complex. The structure suggests the winged-helix domains are not required for incorporation of Dcc1 into the complex, but this should be formally demonstrated.

Other points:

1) The DNA binding activity is relatively weak and required micromolar concentrations of protein and DNA. Is this really a specific binding interaction or just a result of mixing positively and negatively charged macromolecules?

2) Given the weak DNA interactions measured by EMSA, statistics should be given in the ChIP assay. Is there a statistically significant difference in binding for the WH3 deletion of Dcc1 compared to full-length Dcc1? This is important to the conclusion that the WH3 domain is required for chromatin localization.

1st Revision - authors' response

14 December 2016

Comment to both referees: During our revision of the paper, we discovered that that Dcc1WH3 strain originally used for HU-sensitivity and Rad53 phosphorylation experiments (which showed no apparent phenotype) had been erroneously retrieved from storage. We have re-made the strain from scratch, performed these experiments again, and show that the WH3 deletion alone does indeed lead to HU-sensitivity and delayed Rad53 phosphorylation comparable to the other WH deletions (figures 5A and 5B). We have amended the results and discussion to reflect this new data (pages 8,9,11). We have re-checked all other strains employed in this study (figure EV4) and confirmed that this issue was confined solely to the WH3 deletion used for the original checkpoint assays. We sincerely apologise for this error.

Referee #1:

To gain further insight into the role of Ctf18-RFC in different cellular pathways, the authors solved the structure of the two accessory subunits Ctf8 and Dcc1 complexed with a C-terminal fragment of Ctf18. The authors found that the C-terminal domain of Ctf18 interacts through conserved residues with both subunits, Ctf8 and Dcc1. Moreover, they identify three conserved WH domains in Dcc1 that bind DNA. A major success of this work was the subsequent characterization of DNA binding: through EMSA experiments the authors demonstrated that Dcc1 by itself is able to bind to ds and ssDNA through highly conserved residues. It was further demonstrated that loss of Dcc1 or its C-terminal WH domain impaired recruitment of Ctf18-RFC to origins of replication. These findings are novel and important to the field as they bring up interesting questions about the physiological importance of Dcc1 and the mechanism of how Ctf18-RFC recognizes DNA.

The manuscript is easy to read and written in a clear language, and represents a significant advance. I recommend publication in EMBO reports. I do have a few minor critiques, which should be clarified or fixed before publication:

1) On page 9, the authors state "Dcc1 has three consecutive WH domains at its C-terminus which can bind to both ssDNA and dsDNA." However, the in vitro data only show a role for WH3 in direct DNA binding. Although WH1&2 are important for efficient recruitment to chromatin at replication forks, this could be through other means than direct binding to DNA. The above statement needs clarification.

We have amended the text here to make it clear that only the third WH3 is capable of binding DNA.

2) Also on page 9, the authors state "Like the transcription factors, RFC-Ctf18 uses this domain as a bridge between a large complex and WH domains able to bind DNA." But they also claim that "To our knowledge, this is a completely novel organization of WH domains." These two statements seem to be in conflict with each other.

We agree that the wording here is ambiguous. Our point was that the consecutive triple-WH domain architecture is novel (as far as we know), but that the general organisation of the protein complexes (i.e. WH domains linked via a beta-barrel dimerisation interface to a larger complex) is utilised elsewhere. We have re-written this section to make this clearer.

3) On page 10: "which is subsequently capable of binding Pol2." This implies an sequential order of binding, but there is no data to support this. This statement needs to be clarified.

We have re-written this sentence to reflect this.

4) Figure 6 shows the collar region of the clamp loader as an open ring, but it is a closed disk in all known clamp loaders (Bowman et al 2006, Simonetta et al 2009, Kelch et al 2011).

We have re-drawn this figure to make clear that the collar is closed.

5) The nomenclature for the C-terminal fragment of Ctf18 is not consistent throughout the ms.

We have changed all nomenclature to iCtf18C and defined what this is in the results section.

6) No figure calls for Fig. 3B and C.

We have altered the main text referring to the figure to ensure it refers to all panels.

7) The K364A mutation appears to have a rather weak effect on binding and it is not clear whether this should be included in the set of mutations that inhibit binding.

It is a weak effect, though it appears to be real. We have changed the text to mention this point.

It is this reviewer's policy to review non-anonymously when possible. (I don't want to write reviews that I would feel embarrassed about my tone or content if the authors were to actually know my identity.) If the editor and/or authors want clarification about these reviews, please contact: Brian Kelch PhD, UMass Medical School (brian.kelch@umassmed.edu)

Referee #2:

The manuscript by Wade et al. describes the first structure of the Dcc1 and Ctf8 subunits of the RFC-Ctf18 complex. There are four RFC complexes in eukaryotic cells that serve as clamp loaders that load (and unload) ring-shaped sliding clamps on DNA. These RFC complexes contain four "small" subunits in common, but differ in the large subunit, which gives each RFC complex a different cellular function. As a group, the RFC complexes are essential to DNA replication, maintaining genome integrity, sister chromatid cohesion, and activating cellular checkpoints. RFC-Ctf18 differs from the other RFC complexes in that it contains two small accessory subunits, Dcc1 and Ctf8. Given the critical importance of the RFC complexes to genome maintenance and the many unanswered questions regarding how the differences in subunit composition give the RFC

complexes different cellular functions, this manuscript is highly significant.

Surprisingly, the structure revealed that Dcc1 contains three winged-helix domains that often function as DNA binding and/or protein interaction domains. Dcc1 and Ctf8 also interact through a domain that is reminiscent of RNase H2 and a number of transcription factors. These structural features suggest the Dcc1 protein may be important for DNA binding and localizing RFC-Ctf18 to chromatin for its function. The authors perform experiments to test these hypotheses. These functional assays add another dimension to the paper and increase the impact of their structural data.

Because the biochemical and cellular experiments presented to tease out the functions of Dcc1 are important to this paper, there are a couple of key controls that are missing and need to be done to properly interpret the results. First, the authors must show that the Dcc1 deletion mutants are expressed at levels similar to full-length Dcc1. This is important for all of the experiments in Figure 5 (measuring ChIP relative to input in Fig. 5D does not address this). Second, the authors should show that Dcc1 deletion mutants are assembled into the Rfc1-Ctf18 complex. The structure suggests the winged-helix domains are not required for incorporation of Dcc1 into the complex, but this should be formally demonstrated.

We have performed Westerns probing for Dcc1 in the strains employed for all the in vivo studies (Figure EV4A and EV4C). These confirm expression of the truncated mutants at levels similar to the wild-type. We have purified recombinant Dcc1-Ctf8-Ctf18C complexes using the WH3 and WH2-3 deletions (Figure EV4B) confirming assembly of the complexes. WH1-3 could not be expressed in E. coli, but the presence of the protein in cell extracts suggests that it is incorporated in the complex, as free Dcc1 is unstable. The points have been incorporated into the main text results section. We have removed the gel filtration trace presented in the previous figure S3C as this information is effectively superseded by the new figure EV4B.

Other points:

1) The DNA binding activity is relatively weak and required micromolar concentrations of protein and DNA. Is this really a specific binding interaction or just a result of mixing positively and negatively charged macromolecules?

This is a good point. We agree that the observed DNA affinity is somewhat low, but the fact that three point mutants in conserved residues are sufficient to totally eliminate DNA binding does suggest that some specificity is present. The conservation of the putative DNA-binding surface also argues a specific function (figure 3C). Given the Ctf18 complex also interacts with DNA via the RFC subunits, and probably also via a pol-epsilon mediated link, the relatively low affinity of the WH domains may not be so surprising.

2) Given the weak DNA interactions measured by EMSA, statistics should be given in the ChIP assay. Is there a statistically significant difference in binding for the WH3 deletion of Dcc1 compared to full-length Dcc1? This is important to the conclusion that the WH3 domain is required for chromatin localization.

We have performed a three-way ANOVA analysis of the ChIP data. The differences between the wild-type and either dcc1 deletion, or dcc1 WH3 deletion were aggregated across all three early-firing origins, and found to be statistically significant. This information has been included in the legend to figure 5.

2nd Editorial Decision

04 January 2017

Thank you for the submission of your revised manuscript to EMBO reports. We have now received the full set of referee reports that is copied below. As you will see the referees are very positive about the study and suggest only minor revisions to two figures. Apart from this, there are a few things that we need from the editorial side before we can proceed with the official acceptance of your study:

- You have submitted information about the yeast strains used in the study as supplemental material (Appendix Table S2). Please move this table into the Materials & Methods section, as all information on materials and methods must be part of the main manuscript.

- Information on data quantification appears to be missing for Figure 5C. Can you please specify the number "n" for how many experiments were performed and the error bars (e.g. SEM, SD) in the respective figure legend?

- Please provide the accession codes for the protein structures deposited in the Protein Data Bank.

- Please provide a completed 'Author checklist', which covers animal welfare, human subjects, data deposition and ethics. The Author checklist is also available for download on our homepage (Author Guidelines). Please note that the filled form will be published as part of the Review Process File.

We look forward to seeing a final version of your manuscript as soon as possible. Please let me know if you have questions or comments regarding the revision.

REFeree REPORTS

Referee #1:

The authors have suitably revised the manuscript, which I believe should be published in EMBO Reports. This work represents a significant advance in the field, and should be quickly shared with the greater community. Two (very) minor points:

1) In Figure 6, it is not clear what the '(5)' means next to RFCctf18. Is this to represent the 5-subunit core complex? Perhaps 'core complex' would be more clear to the casual reader.

2) It would be good to include in the legend of Figure EV3 panel B that those EMSA shifts were performed with full length Dcc1, rather than the truncations used in EV3 panel A.

Brian Kelch, UMass Med School

Referee #2:

The authors did a good job in addressing all of the criticisms raised in the initial review, and the manuscript is suitable for publication. This is a nice piece of work that will be well-received by the readers of EMBO Reports.

2nd Revision - authors' response

05 January 2017

Responses to Editorial Issues and Outstanding Reviewer's Comments

- You have submitted information about the yeast strains used in the study as supplemental material (Appendix Table S2). Please move this table into the Materials & Methods section, as all information on materials and methods must be part of the main manuscript.

This has been moved into material and methods (page 14, table 1) and removed from the appendix.

- Information on data quantification appears to be missing for Figure 5C. Can you please specify the number "n" for how many experiments were performed and the error bars (e.g. SEM, SD) in the respective figure legend?

For these experiments, n=3 and the error bars present the SD. Included in figure legend on page 22.

- Please provide the accession codes for the protein structures deposited in the Protein Data Bank.

We have submitted the structural data to the PDB under accession codes 5MSM and 5MSN. This information has been included at the end of the manuscript (page 15).

- Please provide a completed 'Author checklist', which covers animal welfare, human subjects, data deposition and ethics. The Author checklist is also available for download on our homepage (Author Guidelines). Please note that the filled form will be published as part of the Review Process File.

This has been completed.

Additional Reviewer Queries:

1) In Figure 6, it is not clear what the '(5)' means next to RFCctf18. Is this to represent the 5-subunit core complex? Perhaps 'core complex' would be more clear to the casual reader.

Revised in Figure 6.

2) It would be good to include in the legend of Figure EV3 panel B that those EMSA shifts were performed with full length Dcc1, rather than the truncations used in EV3 panel A.

Included in figure legend. The actual protein used for the DNA binding experiments was the crystallised construct lacking the first 90 residues as described in the main text. This has been clarified in the legend for figure 3 as well as EV3.

3rd Editorial Decision

09 January 2017

I am very pleased to accept your manuscript for publication in the next available issue of EMBO reports. Thank you for your contribution to our journal.

YOU MUST COMPLETE ALL CELLS WITH A PINK BACKGROUND

Corresponding Author Name: Martin R. Singleton

Manuscript Number: EMBOR-2016-42825V3